# Orf Virus IL-10 and VEGF-E Act Synergistically to Enhance Healing of Cutaneous Wounds in Mice

**DOI:** 10.3390/jcm9041085

**Published:** 2020-04-11

**Authors:** Lyn M. Wise, Gabriella S. Stuart, Nicola C. Jones, Stephen B. Fleming, Andrew A. Mercer

**Affiliations:** 1Department of Pharmacology and Toxicology, School of Biomedical Sciences, University of Otago, Dunedin 9054, New Zealand; gabriella.stuart@otago.ac.nz (G.S.S.); nicky.jones@otago.ac.nz (N.C.J.); 2Department of Microbiology and Immunology, School of Biomedical Sciences, University of Otago, Dunedin 9054, New Zealand; stephen.fleming@otago.ac.nz (S.B.F.); andy.mercer@otago.ac.nz (A.A.M.)

**Keywords:** Orf virus, VEGF-E, IL-10, skin, wound, healing, scarring

## Abstract

Orf virus (OV) is a zoonotic parapoxvirus that causes highly proliferative skin lesions which resolve with minimal inflammation and scarring. OV encodes two immunomodulators, vascular endothelial growth factor (VEGF)-E and interleukin-10 (ovIL-10), which individually modulate skin repair and inflammation. This study examined the effects of the VEGF-E and ovIL-10 combination on healing processes in a murine wound model. Treatments with viral proteins, individually and in combination, were compared to a mammalian VEGF-A and IL-10 combination. Wound biopsies were harvested to measure re-epithelialisation and scarring (histology), inflammation, fibrosis and angiogenesis (immunofluorescence), and gene expression (quantitative polymerase chain reaction). VEGF-E and ovIL-10 showed additive effects on wound closure and re-epithelialisation, and suppressed M1 macrophage and myofibroblast infiltration, while allowing M2 macrophage recruitment. The viral combination also increased endothelial cell density and pericyte coverage, and improved collagen deposition while reducing the scar area. The mammalian combination showed equivalent effects on wound closure, re-epithelialisation and fibrosis, but did not promote blood vessel stabilisation or collagen remodeling. The combination treatments also differentially altered the expression of transforming growth factor beta isoforms, *Tgfβ1* and *Tgfβ3.* These findings show that the OV proteins synergistically enhance skin repair, and act in a complimentary fashion to improve scar quality.

## 1. Introduction

Orf virus (OV) is the type species of the *Parapoxvirus* genus of the family *Poxviridae* [1]. This virus infects skin wounds in ungulates and humans, causing lesions that are remarkable for their extensive epidermal hyperplasia and blood vessel formation [2,3]. Despite inducing such proliferative lesions, OV infection causes only a transient increase in skin inflammation, with impairment of innate immune cell trafficking [4,5,6]. Intriguingly, even the hyper-proliferative lesions reported in immune-compromised hosts resolve with minimal inflammation and scarring [7]. These features of OV lesions suggest that the virus manipulates the wound healing response of the skin.

Wound healing is a complex process involving three overlapping but interconnected phases: inflammation, proliferation and remodeling [8]. Following clot formation, inflammatory and innate immune cells enter the wound site. These cells produce soluble mediators, such as growth factors and cytokines, which direct the proliferation and migration of keratinocytes to re-epithelialise the wound surface, and fibroblasts and blood vessels to form the new granulation tissue. The soluble mediators also stimulate the differentiation of fibroblasts into myofibroblasts, leading to wound contraction. The fibroblasts direct the remodeling process as well, producing the collagen matrix that forms the residual scar tissue. The three phases are tightly regulated by the timing of the expression and the abundance of growth factors and cytokines [9]. Dysregulation of these mediators can cause the healing process to stall in the inflammatory phase, resulting in wounds that fail to heal, or to remain in the proliferative phase, which leads to impaired remodeling and the formation of hypertrophic or keloid scars.

The proliferative nature of OV lesions is due to the expression of a viral growth factor, a homologue of vascular endothelial growth factor (VEGF) [3,10,11]. VEGF-A, the prototype and most studied member of the mammalian VEGF family, promotes wound re-vascularisation and re-epithelialisation through its interactions with the VEGF receptor (VEGFR)-2 [12]. However, through its interactions with VEGFR-1, VEGF-A also contributes to wound inflammation and oedema [13], detrimentally influencing scar formation [14]. The viral VEGF homologue, designated VEGF-E, like VEGF-A, interacts with VEGFR-2 promoting wound re-epithelialisation [15,16,17] and blood vessel formation in intact and wounded skin [15,18,19,20,21]. VEGF-E, differs from VEGF-A, as it does not bind VEGFR-1, which mediates the recruitment of inflammatory monocytes [21,22,23]. VEGF-E also exerts immunomodulatory effects, inducing the expression of anti-inflammatory cytokine interleukin (IL)-10, and limiting inflammation and fibrosis in wounded skin [17,24].

The transient inflammatory and immune response to OV infection has been attributed, in part, to the expression of a viral cytokine, a homologue of IL-10 [1]. In mammals, IL-10 plays a major role in the limitation and termination of wound inflammation, and its anti-fibrotic effects contribute to scar-free healing [25]. The viral protein, ovIL-10, has equivalent immunosuppressive activity to mammalian IL-10, reducing pro-inflammatory cytokine production [26,27,28,29] and limiting the recruitment of monocytes, dendritic cells and mast cells to inflamed skin [30]. When applied to wounded skin, the ovIL-10 protein dampened macrophage infiltration and fibrosis, reducing scar tissue formation to an equivalent extent to the mammalian IL-10 [29]. 

Given the successful resolution of OV lesions following the concurrent expression of VEGF-E and ovIL-10, we hypothesise that, in the absence of OV infection, this combination of proteins may improve skin repair to a greater extent than the individual proteins. This study therefore examined the combined effects of VEGF-E and ovIL-10 on healing and scarring processes in a murine full-thickness cutaneous wound model. These effects were also compared to those of the equivalent combination of murine proteins, VEGF-A and IL-10. The dosing schedule chosen mimicked the expected timing of IL-10 and VEGF-E expression during viral infection, while the amount of each protein administered was chosen based on their reported effects in ovine and murine skin [11,16,29,30]. A subcutaneous route of administration was also chosen, as its efficacy had been demonstrated with the individual proteins [16,17,29]. The findings from this study suggest that the combination of VEGF-E and ovIL-10 exerts synergistic and complimentary effects on wound healing responses. While both the mammalian and viral combination increased skin repair, the viral combination of VEGF-E and ovIL-10 showed the greatest reduction in scar formation.

## 2. Experimental Section

### 2.1. Recombinant Proteins 

Recombinant FLAG-tagged murine VEGF-A isoform_164_ and IL-10, and viral VEGF-E and ovIL-10 were expressed in HEK293-EBNA cells, purified and quantitated as previously described [20,27].

### 2.2. Ethics Statement 

Animal procedures were conducted following approval from the Animal Ethics Committee of the University of Otago (#67/08) dated 11 December 2008. Animals were sourced and housed in the Hercus-Taieri Resource Unit (Dunedin, New Zealand).

### 2.3. Full-Thickness Skin Wound Model 

C57BL/6 mice (female, 8 weeks of age) were anaesthetised and shaved, then two full-thickness excisional wounds were made on the dorsal skin using a sterile 4 mm biopsy punch [17]. Animals were separated into five groups, control (vehicle only), VEGF-E-treated, ovIL-10-treated, VEGF-E and ovIL-10-treated and VEGF-A and IL-10-treated. Day 2 post-wounding, each wound was treated with 50 μL subcutaneous (SC) injection of vehicle (phosphate-buffered saline (PBS)) or VEGF-E (1 µg), ovIL-10 (100 ng), VEGF-E and ovIL-10 (1 µg/100 ng) or VEGF-A and IL-10 (1 µg/100 ng) diluted in PBS. The initial injection was administered 0.5 cm from the lateral edge, aiming towards the wound. Repeat treatments administered on days 4, 6 and 8, were applied to the caudal, medial, then cranial wound edges, respectively. Digital photographs were captured with a ruler aligned next to the wound while mice were immobilised with inhaled isoflurane (1%–5%). The wound size was quantified using ImageJ (http://imagej.nih.gov/ij/), with changes expressed as a percentage of the original wound size. Mice were euthanised on day 3, 6, 9 and 16, wounds surgically excised, then bisected at the centre of the wound along the medial–lateral axis. The bisected skin was fixed in 0.5% zinc salts solution, processed into paraffin wax then sectioned (4 μm) or stored in RNA*later* RNA Stabilisation solution (Qiagen, Hilden, Germany).

### 2.4. Histology

Paraffin-embedded skin sections (4 μm) were stained with Martius scarlet blue then visualised under bright light, or with picrosirius red visualised under a dark-field with a polarised filter [31]. Images were taken of the section at 10X magnification then converted into panoramas using Photoshop (Adobe Systems, San Jose, CA, USA). 

Using ImageJ, measurements of the wound width, between the cut ends of the panniculus carnosus, and the proportion covered with neo-epidermis, were taken in images of MSB-stained skin harvested at day 3 post-wounding. Re-epithelialisation was calculated as a percentage of the wound width covered by neo-epidermis [16,29]. From images of MSB-stained sections of skin harvested day 16 post-wounding, three epidermal thickness measurements within the cut edge of the panniculus carnosis were taken using ImageJ, and the average of these measurements calculated [32]. Fibrotic scar tissue extending to the depth of the panniculus carnosis was outlined and measured. 

Collagen density within the scar tissue was examined in images of the picrosirius red-stained sections from skin harvested at day 16 post-wounding [17]. Images were converted to 8 bit in ImageJ, scar tissue was outlined, the threshold adjusted automatically to highlight the stained collagen, then measurements were taken as a percentage of the total scar tissue area. Collagen complexity was examined in images of picrosirius red-stained sections from skin harvested at day 16 post-wounding taken at 40× magnification. Fast Fourier transform (FFT) statistical analysis was conducted on the images using ImageJ. The collagen orientation index was determined by calculating the width/length ratio of the zero-order maximum in the generated plot of the image [31]. In normal skin, the collagen bundles form a basket weave-like pattern that results in a circular power spectrum and an orientation index approaching 1. In mature scar tissue, however, the collagen bundles tend to be parallel in nature, leading to an elongated power spectrum and a small orientation index.

### 2.5. Immunohistochemistry

Paraffin-embedded skin sections (4 μm) were incubated with antibodies to F4/80 (AlexaFluor^®^ 488-BM8; #14-4801, eBioscience, San Diego, CA, USA; dilution 1:100, or AlexaFluor^®^ 594-BM8; #123140, Biolegend, San Diego, CA; dilution 1:100), CD206 (#ab64693, Abcam; dilution 1:200) followed by incubation with AlexaFluor^®^ 488-conjugated antibody (#A-11008, Invitrogen, Carlsbad, CA, USA; 1:500 dilution), calprotectin (fluorescein isothiocyanate-MAC387, #ab7492, Abcam, Cambridge, England, dilution 1:200), inducible nitric oxide synthase (iNOS; #ab3523, Abcam; dilution 1:100) with AlexaFluor^®^ 594-conjugated antibody (#A-11037, Invitrogen; dilution 1:500), CD31 (#ab28364, Abcam; dilution 1:200) or von Willebrand factor (vWF; #A 0082, DakoCytomation, Glostrup, Denmark; dilution 1:200) with AlexaFluor^®^ 488-conjugated antibody, alpha smooth muscle actin (αSMA; Cy3-1A4; #C6198, Sigma-Aldrich, St Louis, MO, USA; dilution 1:400), or vimentin (AlexaFluor^®^ 488-D21H3; #5741, Cell Signaling Technology, Danvers, MA; dilution 1:100). Nuclei were stained with DAPI (#D3571, Invitrogen, 75 nM) and slides mounted with SlowFade Gold (Invitrogen). Images were taken of the stained section at 40× magnification, then converted into panoramas using Photoshop.

Macrophages were quantitated in images taken of F4/80-stained sections from skin harvested at day 3, 6, and 9 post-wounding. The wound margin and bed to a point 500 µm distal to the cut edge of the panniculus carnosus was outlined in ImageJ, then green-stained cells were counted manually [17,29]. The outlined region was measured, and cell number expressed relative to that area. M1 and M2 macrophages were assessed in images taken of calprotectin and iNOS- and CD206 and F4/80-stained sections, respectively [33,34], with cells co-stained green and red counted manually, and cell number was expressed relative to the corresponding area measurement. 

Myofibroblasts were quantitated in images taken of vimentin and αSMA-stained sections from skin harvested at day 6 post-wounding [17,29]. The wound margin and granulation tissue were outlined in ImageJ, then the number of green and red co-stained cells were counted manually. The corresponding region was measured, and cell number was expressed relative to that area.

Blood vessels were quantitated in images taken of CD31 and αSMA-stained sections, and vWF and αSMA-stained sections, from skin harvested at day 6 and 9 post-wounding, respectively [15,17]. The wound margin and granulation tissue were outlined in ImageJ, then the number of green-stained CD31 or vWF^+ve^ endothelial cells, and blood vessels surrounded by red-stained αSMA^+ve^ pericytes, were counted manually. The corresponding region was measured, with the cell and blood vessel numbers expressed relative to that area.

### 2.6. Quantitative PCR

Four treatment-matched wounds were combined per sample. Total RNA was then extracted using Trizol (Invitrogen) and Proteinase K (Sigma-Aldrich) digestion then purified using the RNeasy^®^ Mini Kit (Qiagen) [29]. Total RNA was reverse transcribed using Superscript III (Invitrogen), oligo(dT)_15_ and random hexamer primers. Real-time quantitative PCR was conducted with gene specific primers (glyceraldehyde 3-phosphate dehydrogenase (*Gapdh*) 5′ CAAAAGGGTCATCATCTCCG 3′ and 5′ TAAGCAGTTGGTGGTGCAGGA 3′; matrix metalloproteinase (*Mmp*)2 5′ ACACCTACACCAAGAACTTCCG 3′ and 5′ GGGCCAGTACCAGTGTCAGTATC 3′; *Mmp9* 5′ CGGACATTGTCATCCAGTTTGG 3′ and 5′ AATGGGCATCTCCCTGAACG 3′; macrophage inflammatory protein (*Mip*)*-2* 5′ CATCCAGAGCTTGAGTGTGACG 3′ and 5′ GCTTCAGGGTCAAGGCAAACTT 3′; *Il-1β* 5′ CTTCCAGGATGAGGACATGAGC 3′ and 5′ AGTGCAGTTGTCTAATGGGAACG 3′; *Il-10* 5′ CTAGAGCTGCGGACTGCCTTC 3′ and 5′ AGGAGTCGGTTAGCAGTATGTTG 3′; *Vegf-a* 5′ GCAGGCTGCTGTAACGATGAAG 3′ and 5′ GCTTTGGTGAGGTTTGATCCG 3′; transforming growth factor (*Tgf*)*β*1 5′ GCTTGCAGAGATTAAAATCAAGTG 3′ and 5′ ACCAAGGTAACGCCAGGAATTG 3′ *Tgfβ3* 5′ ATTCGACATGATCCAGGGAC 3′ and 5′ TCTCCACTGAGGACACATTGA 3′) and SYBR^®^ Green PCR Master Mix (Invitrogen) using the ABI PRISM 7700 Sequence Detection System (Applied Biosystems, Foster City, CA, USA). All mRNA levels were quantitated relative to *Gapdh* and unwounded skin taken from age and sex-matched mice [16,17,29]. The post-wounding changes in the *TGFβ1* and *TGFβ3* mRNA levels at each time point were also calculated as a ratio. 

### 2.7. Statistical Analyses

Values were obtained for each wound group (*n* = 8), and the mean ± SEM calculated as required. The normality of each dataset was confirmed using the Shapiro–Wilk normality test. Single or two-factor ANOVA was then conducted with significant points of difference between means determined using Sidak’s test, following correction for multiple comparisons. Values of *p* ≤ 0.05 were considered statistically significant.

## 3. Results

### 3.1. VEGF-E and ovIL-10 Synergistically Enhance Wound Closure and Re-epithelialisation

The effects of viral VEGF-E (1 µg) and ovIL-10 (100 ng), individually and in combination, on tissue repair, were examined in a mouse model of cutaneous wound healing, and compared to the effects of an equivalent combination of murine VEGF-A (1 µg) and IL-10 (100 ng). Photographs taken during wound healing revealed macroscopic differences in healing kinetics between the treatment groups (Figure 1a). At the point of the first treatments by SC injection, the wound area was equivalent across treatment groups (Figure 1b, *p* ≥ 0.05). However, by day 4, treatment with VEGF-E and ovIL-10 had resulted in a significant reduction in wound size compared with mock-treated wounds (Figure 1b, *p* ≤ 0.0001). Wounds treated with the combination of VEGF-E and ovIL-10 were also significantly smaller in area at day 4 than wounds that received treatment of either VEGF-E (*p* = 0.05) or ovIL-10 (*p* = 0.01). To determine if this synergistic effect was specific to the viral proteins, wounds were treated with the equivalent combination of their murine homologues, VEGF-A and IL-10. The administration of VEGF-A and IL-10 in combination led to a reduction in wound area at day 4 that was equivalent to that of the viral protein combination (Figure 1b, *p* ≥ 0.05). At subsequent time points, many wounds appeared to have closed, and no measurable differences in area were evident between treatment groups (Figure 1b). 

As changes in wound closure were observed following a single protein treatment, re-epithelialisation of the wound was examined histologically at day 3 post-wounding. At this time-point, the neo-epidermis was observed, projecting down from the wound edge, between the wound bed and the fibrin clot (Figure 2a,b). VEGF-E treatment led to a 2-fold increase in re-epithelialisation over that of mock-treated wounds (Figure 2b,c, *p* < 0.0001). Treatment with ovIL-10 also increased wound re-epithelialisation by 1.5-fold (Figure 2b,c, *p* = 0.01) within the same timeframe. Wounds treated with the combination of VEGF-E and ovIL-10 showed a more substantial 2.5-fold increase in re-epithelialisation (Figure 2b,c, *p* < 0.0001) compared to control wounds, the level of which was significantly greater than individual treatment with VEGF-E (*p* = 0.004) or ovIL-10 (*p* < 0.002). The administration of VEGF-A and IL-10 in combination led to an increase in wound re-epithelialisation that was equivalent to that of the viral protein combination, 2.5-fold over that of control wounds (Figure 2b,c, *p* < 0.0001). Consistent with the observations relating to wound closure, the majority of wounds in each treatment group were re-epithelialised by day 6 post-wounding (Appendix A).

### 3.2. VEGF-E and ovIL-10 have Complimentary Effects on the Response of Macrophages to Wounding

The macrophage infiltrate within the wounded skin was examined at day 3, 6 and 9 post-wounding, after one, two and four protein treatments, respectively. It had previously been shown in this wound model that the peak macrophage infiltration occurred at day 6 [29]. 

An accumulation of F4/80^+ve^ macrophages was evident at the margin of vehicle-treated control wounds at day 6 (Figure 3a,b), which persisted until day 9 post-wounding (Figure 3c). The number of macrophages in the wound bed and margin was 1.6-fold lower in VEGF-E-treated wounds at day 6 (Figure 3b,c, *p* = 0.004), but was equivalent to that of control skin at day 9. A significant 3-fold reduction in macrophages was observed at both day 6 and 9 in ovIL-10-treated skin and in wounds treated with VEGF- E and ovIL-10 (Figure 3b,c, *p* = 0.001 and *p* = 0.03, respectively). The macrophage number was also reduced by 2-fold at day 6 following treatment with VEGF-A and IL-10 (Figure 3b,c, *p* = 0.0001), but this reduction was not maintained relative to controls at day 9 post-wounding. 

Wound macrophages exhibit different phenotypes as a wound matures, with M1 and M2 macrophages acting in a pro-inflammatory or anti-inflammatory and reparative manner, respectively. As the viral protein treatments suppressed macrophage numbers to the greatest extent at day 6, the phenotype of the macrophages present in the wounds was examined at this point. Staining for calprotectin and iNOS, which are expressed in M1 macrophages, showed double-stained cells at the margin of mock-treated wounds after 6 days (Figure 3d), but these cells were less abundant than those staining for F4/80. Staining for CD206, a marker of M2 macrophages, however, was evident on approximately three-quarters of the F4/80^+ve^ cells found within the wound bed and margin of control skin at this time-point (Figure 3e). 

The number of calprotectin^+ve^ iNOS^+ve^ M1 macrophages in the wound bed and margin was significantly lower in protein-treated skin at day 6, relative to that observed in mock-treated wounds (2–3.5-fold, Figure 3d,f, *p* ≤ 0.03). A 2-fold reduction in CD206^+ve^ F4/80^+ve^ M2 macrophage number was also observed in ovIL-10-treated skin (Figure 3e,g, *p* = 0.02), and in wounds treated with VEGF-A and IL-10 (Figure 3e,g, *p* = 0.05). By contrast, the M2 macrophage number in skin wounds treated with VEGF-E, with and without ovIL-10, was equivalent to controls (Figure 3e,g).

### 3.3. ovIL-10, with and without VEGF-E, Reduces the Response of Myofibroblasts to Wounding 

The presence of myofibroblasts within the wounded skin was examined at day 6 post-wounding, after two protein treatments. This time point was chosen as it had previously been shown in this wound model to be the peak for myofibroblast infiltration [29]. 

A substantial amount of overlapping vimentin and αSMA co-staining, consistent with myofibroblasts, was observed in the wound margin of control wounds (Figure 4a,b). In wounds treated with ovIL-10, the number of co-stained cells within the wound margin and bed was significantly reduced relative to control wounds (2-fold, Figure 4b,c, *p* = 0.0006). A similar reduction in myofibroblast staining, compared to control wounds, was observed in wounds treated with VEGF-E and ovIL-10 (Figure 4b,c, *p* = 0.007). However, no change was observed in VEGF-E- or VEGF-A and IL-10-treated wounds (Figure 4b,c). 

### 3.4. VEGF-E and ovIL-10 Synergistically Enhance Wound Re-Vascularisation

Following wound closure, at day 6 and 9 post-wounding, after two and four protein treatments, re-vascularisation of the wound bed was examined. These days were chosen as the time points had previously been shown in this wound model to be the peak for wound re-vascularisation [29]. 

At day 6, the newly formed granulation tissue and adjacent wound margin contained a large of number of individual CD31^+ve^ endothelial cells, some of which were surrounded by αSMA^+ve^ cells consistent with pericytes (Figure 5a,b). VEGF-E treatment led to a 2-fold increase in the number of endothelial cells within the granulation tissue and adjacent wound margin compared to control wounds at this time point (Figure 5b,c, *p* = 0.03). Treatment with ovIL-10, however, had little effect on endothelial cell number (Figure 5b,c). Wounds treated with the combination of VEGF-E and ovIL-10 also showed an increased endothelial cell number compared to mock-treated wounds (4-fold, Figure 5b,c, *p* < 0.0001), with levels significantly greater than that of wounds treated with either VEGF-E or ovIL-10 (*p* < 0.0001). Treatment with VEGF-A and IL-10 also increased the number of endothelial cells (2-fold, Figure 5b,c, *p* < 0.01), but only to a level equivalent to the VEGF-E treatment.

At day 9, endothelial cell expression of CD31 was diminished relative to day 6, while that of vWF was increased (not shown). The wound edge and granulation tissue at that time point contained many vWF^+ve^ endothelial cells, a substantial proportion of which were surrounded by αSMA^+ve^ cells, consistent with mature blood vessels (Figure 5d,e). Avascular αSMA^+ve^ cells, consistent with myofibroblasts, were also observed, but were not included in the analyses. VEGF-E treatment led to a 2-fold increase in mature blood vessels at day 9 post-wounding compared to control wounds (Figure 5e,f, *p* = 0.0008), while a modest increase was observed following treatment with ovIL-10. Treatment with VEGF-A and IL-10 also led to a 2-fold increase in pericyte-coated blood vessels (Figure 5e,f, *p* = 0.003). The greatest density of mature blood vessels was observed with the combination of VEGF-E and ovIL-10 (2.5-fold increase relative to mock-treated wounds, Figure 5e,f, *p* < 0.0001), and this was significantly greater than the individual VEGF-E and ovIL-10 treatments (Figure 5e,f, *p* = 0.0005 and *p* < 0.0001, respectively), and the combination treatment of VEGF-A and IL-10 (Figure 5e,f, *p* = 0.001). Attempts to confirm the pericyte phenotype of αSMA^+ve^ cells surrounding the blood vessels, through staining for NG2 chondroitin sulfate proteoglycan and platelet derived growth factor (PDGF)-β receptor, were unsuccessful, as the zinc-fixed sections used in this study did not withstand the temperature required for retrieval of these antigens (>95 °C).

### 3.5. VEGF-E and ovIL-10 have Complimentary Effects on Scar Size and Quality

The size and quality of the newly formed scar was examined at day 16 post-wounding, eight days after the final protein treatment. Day 16 was chosen for these analyses as it had previously been shown in this wound model that wound contraction and granulation tissue formation have resolved and scar tissue/collagen deposition has begun by this time point [29].

A narrow scar was evident at day 16, with a thickened neo-epidermis and collagen-rich scar tissue extending to the depth of the panniculus carnosus (Figure 6a). The neo-epidermis in wounds treated with the vehicle was 2-fold thicker than the neo-epidermis in wounds of any of the treatment groups (Figure 6a,b, *p* ≤ 0.01). The dermal scar was 35% smaller in wounds treated with VEGF-E, ovIL-10, and VEGF-E and ovIL-10, compared to mock-treated wounds (Figure 6a,c, *p* ≤ 0.04). The dermal scar was not, however, reduced in wounds treated with VEGF-A and IL-10 (Figure 6c) compared to controls, but was significantly larger than in wounds treated with VEGF-E, ovIL-10, and VEGF-E and ovIL-10 (Figure 6a, c, *p* ≤ 0.05). 

The deposition of collagen and formation of a basket-weave matrix was evident in the scar tissue (Figure 6d). VEGF-E- and VEGF-E and ovIL-10-treated wounds showed a significant 20%–22% increase in the amount of collagen staining within the scar tissue compared with control wounds (Figure 6d,e, *p* = 0.0004 and *p* = 0.005, respectively). No increase in collagen staining was observed following treatment with ovIL-10 or VEGF-A and IL-10 (Figure 6d,e). FFT analysis showed that the orientation of collagen bundles within the scar tissue from VEGF-E and VEGF-E and ovIL-10-treated wounds had larger aspect ratios than control wounds (1.4-fold, Figure 6f, *p* = 0.02 and *p* = 0.003, respectively), which was consistent with observations that the collagen bundles were forming a random basket-weave pattern closer to that of normal skin (Figure 6d). By contrast, the FFT aspect ratio of scar tissue from VEGF-A and IL-10-treated wounds was below that of control wounds, and significantly lower than wounds treated with the viral protein combination (1.8-fold, Figure 6f, *p* = 0.0003), due to the presence of bundles that were more parallel in orientation (Figure 6d).

### 3.6. VEGF-E and ovIL-10 Alter the Genetic Expression of Key Wound Healing Mediators

Following treatment with the viral and mammalian protein combinations, mRNA levels of key wound mediators were examined on day 3, 6, 9 and 16 post-wounding using quantitative PCR. 

The administration of VEGF-E and ovIL-10 enhanced expression of matrix metalloproteinases which modulate wound re-epithelialisation, with *Mmp2* and *Mmp9* mRNA levels significantly increased on day 3, 6 and 9 (Figure 7a,b; *p* ≤ 0.05). Compared to wounds treated with the viral proteins, VEGF-A and IL-10-treated wounds had a reduced level of *Mmp2* mRNA at day 3 (Figure 7a, *p* = 0.04), while *Mmp9* mRNA levels were lower at day 3 and 9 (*p* = 0.003 and *p* = 0.01, respectively). 

VEGF-E and ovIL-10 treatment initially decreased mRNA levels for the inflammatory mediators, *Il-1β* and *Mip-2* on day 3 (Figure 7c–d, *p* < 0.0001), but levels were then greater than controls at day 6 (*p* < 0.0001). Compared to wounds treated with viral proteins, VEGF-A and IL-10-treated wounds had higher levels of *Il-1β* at day 3 and 6 (Figure 7c, *p* < 0.0001), and *Mip-2* at day 3 (Figure 7d, *p* = 0.003).

The protein combinations also altered expression of endogenous *Il-10* and *Vegf-a* in the wound. The level of *Il-10* mRNA was higher in VEGF-E and ovIL-10 treated wounds than controls at day 3 (Figure 7e, *p* < 0.0001). Wounds treated with the viral proteins also had higher *Il-10* mRNA levels than those treated with VEGF-A and IL-10 at day 3 (Figure 7e, *p* < 0.0001). Although *Vegf-a* mRNA levels in the wound were reduced at day 3 by treatment with VEGF-E and ovIL-10 (Figure 7f, *p* < 0.0001), expression levels then increased and were greater than controls at day 6, 9 and 16 (*p* ≤ 0.001). In wounds treated with VEGF-A and IL-10, *Vegf-a* mRNA levels were higher at day 3 than in viral protein-treated wounds (Figure 7f, *p* < 0.0001). 

The treatment of wounds with VEGF-E and ovIL-10 also altered the expression of key regulators of fibrosis, transforming growth factors *Tgfβ1* and *Tgfβ3*. Although *Tgfβ1* mRNA levels were reduced at day 3 by treatment with VEGF-E and ovIL-10 (Figure 7g, *p* = 0.002), expression levels in the wound then increased and were greater than controls at day 6, 9 and 16 (*p* ≤ 0.003). The level of *Tgfβ1* mRNA in wounds treated with the viral proteins was, however, lower than VEGF-A and IL-10-treated wounds on day 3 and 6 (Figure 7g, *p* ≤ 0.0001). *Tgfβ3* mRNA levels were higher in VEGF-E- and ovIL-10-treated wounds than VEGF-A- and IL-10-treated wounds at day 6 (Figure 7h, *p* < 0.0001), and control wounds at day 9 (Figure 7h, *p* < 0.0001). The ratio of *Tgfβ1* mRNA relative to that of *Tgfβ3* mRNA was decreased at day 3 by treatment with VEGF-E and ovIL-10 relative to both VEGF-A- and IL-10-treated and control wounds (Figure 7i, *p* < 0.0001). At day 6, this mRNA ratio was lower in VEGF-E- and ovIL-10-treated wounds than in VEGF-A- and IL-10-treated wounds (Figure 7i, *p* < 0.0001), but was equivalent to control wounds. At the later time points, the *Tgfβ1* mRNA to *Tgfβ3* mRNA ratio was equivalent in mock-treated and VEGF-E- and ovIL-10-treated wounds (Figure 7i).

## 4. Discussion

OV lesions are highly proliferative yet resolve with minimal scarring. These features of OV lesions have been linked to the expression of two viral proteins, VEGF-E and ovIL-10, which independently have been shown to manipulate healing and scarring responses in the skin. This study tested the hypothesis that a combination of these viral proteins would improve healing and limit scarring to a greater extent than the individual proteins. This was tested in a murine full-thickness skin wound model, with the effects of VEGF-E and ovIL-10 individually and in combination, and compared to those of the equivalent combination of murine proteins, VEGF-A and IL-10. The findings supported our hypothesis, as VEGF-E and ovIL-10 acted synergistically to promote wound closure, re-epithelialisation and re-vascularisation, and in a complimentary manner to reduce the size and improve the quality of the residual scar. The mammalian protein combination of VEGF-A and IL-10 also increased skin repair, but without the same beneficial effects on scar formation.

Wound re-epithelialisation occurs as keratinocytes at the wound edge proliferate then migrate across the surface of the wound bed [35]. The cells then differentiate to form the stratified layers that reconstitute the neo-epidermis. In this study, the individual proteins, VEGF-E and ovIL-10, enhanced wound re-epithelialisation, and an additive effect was observed with the viral combination. The increase in re-epithelialisation induced by the viral proteins was also equivalent to that induced by the mammalian combination. The synergistic effects of the viral proteins on wound re-epithelialisation was also consistent with changes in wound size observed at the macroscopic level. Both mammalian and viral VEGF been shown to enhance wound re-epithelialisation in mice [11,16,36,37]. This was attributed to direct mitotic and chemotactic effects on keratinocytes, and the indirect effects of increased MMP2 and MMP9 production on keratinocyte immobilisation and growth factor bioavailability [16,38]. As previously reported with the individual VEGF proteins, increased wound expression of *Mmp2* and *Mmp9* was observed following treatment with the mammalian and viral VEGF and IL-10 combinations. Further analyses are warranted to validate whether changes in mRNA translate to changes in protein abundance, and to ascertain whether those changes occur within the neo-epidermis or elsewhere in the wound.

The role of IL-10 in wound re-epithelialisation is more controversial, as this process is accelerated in mice deficient in IL-10 expression and in those treated with recombination IL-10 [29,39,40]. It is thought that IL-10 may indirectly impact wound re-epithelialisation through modulation of the constitution and timing of the inflammatory infiltrate [41]. However, the injection of IL-10-stimulated macrophages into murine wounds had no effect on wound re-epithelialisation [42]. Another explanation may lie in the ability of IL-10 to induce or suppress the production of keratinocyte-modulating growth factors. Mammalian and viral IL-10s have both been shown to limit the production of TGFβ1 [29,43], which causes cell cycle arrest in keratinocytes [44]. Indeed, treatment with TGFβ1 antagonists early in skin repair has been shown to accelerate re-epithelialisation [45]. The viral and mammalian protein combinations, however, exerted differential effects of *Tgfβ1* mRNA levels. As both treatments accelerated wound re-epithelialisation, the mechanism by which IL-10 synergistically enhances this process, over that of VEGF alone, is unlikely to depend on TGFβ1 levels. IL-10 may also exert proliferative effects on keratinocytes, potentially through Stat3 and Akt activation [46], and the transcriptional regulation of pro-survival Bcl-2 and Cyclin proteins [47].

Macrophages play important roles in skin repair, by cleaning the wounds of bacteria and apoptotic cells, and producing growth factors and cytokines that guide granulation tissue formation and matrix remodeling [48]. Here, the individual and combination VEGF and IL-10 treatments reduced the initial infiltration of macrophages into the wound, and this effect persisted longer with treatments containing the viral IL-10. Both mammalian and viral IL-10 have been reported to reduce macrophage recruitment [29,49,50,51]. Treatment with VEGF-E also reduced the wound macrophage infiltrate, reportedly through increased IL-10 expression [17], and this pattern was retained here following treatment with the viral protein combination. As mammalian IL-10 has been shown to reduce the cell surface expression of intercellular adhesion molecule-1, vascular cell adhesion molecule-1, integrin*β2* and L-selectin [52,53], it is possible that monocyte recruitment to the wound is impeded by these treatments through a loss of adhesion to endothelial cells. To our knowledge, the impact of IL-10 on these cell adhesion molecules has not been studied in the context of a healing wound. Further analysis of macrophage phenotype revealed that all treatments suppressed the recruitment of M1 macrophages. This observation was consistent with the reduced expression of the M1 marker *Il-1β* observed in wounds treated with the viral VEGF-E and ovIL-10 combination, but not in wounds treated with the mammalian combination. Macrophage phenotype is thought to be influenced by VEGFs, with VEGFR-1 being highly expressed in M1 but not M2 macrophages [54,55,56], and VEGF-A reported to promote M2 polarisation *in vitro* [57,58]. However, another study demonstrated that VEGF-A required co-stimulation with IL-10 to support M2 polarisation *in vitro* [59]. Here, the greatest M2 macrophage infiltrate was associated with treatments containing the VEGFR-2-selective VEGF-E, and levels were, for the most part, suppressed by IL-10. Further analyses are therefore needed to dissect the relative contribution of VEGFs, IL-10s and their receptors to the balance between macrophage phenotypes in the wound environment.

Myofibroblasts play important roles in wound contraction and in the deposition of the matrix, processes that are modulated by TGFβ isoforms [60,61]. TGFβ1 induces myofibroblast migration, and their production of αSMA. TGFβ3, by contrast, reduces collagen type I deposition by restricting myofibroblast differentiation. In this study, the viral combination treatment suppressed the accumulation of wound myofibroblasts to a similar extent as treatment with ovIL-10 alone, and to a greater extent than with the mammalian combination treatment. This was consistent with differences in the *Tgfβ1:Tgfβ3* expression ratio between the combination treatments, where a delay in *Tgfβ1* expression was observed following the viral protein treatment. Viral ovIL-10 has been shown to share the anti-fibrotic effects of mammalian IL-10, reducing both *Tgfβ1* and αSMA production [29,43,62,63,64,65]. Pericytes are a major source of myofibroblasts [66], and the maturation of blood vessels is critical to the resolution of fibrosis in the lung and liver [67,68,69]. A stable vasculature may, therefore, counteract myofibroblast differentiation. We observed increased pericyte coverage of the vasculature in wounds treated with VEGF-E, and with the viral and mammalian combinations. However, as only a single marker (αSMA) was used to identify myofibroblasts and pericytes, additional markers are needed to delineate these cell types within the viral protein-treated wounds throughout the healing process. This will facilitate further investigations as to the contributions made by these cells to wound fibrosis and re-vascularisation.

The creation of a blood vessels through angiogenesis is considered essential for successful wound repair, and this process is critically linked to the expression of VEGFs [12]. Consistent with previous reports [17,29], an increased endothelial cell density was observed following treatment with VEGF-E alone, and even more so in combination with ovIL-10. This can be attributed to the direct effects of VEGF-E on endothelial cell proliferation and migration observed *in vitro* and *in vivo* [11,16,18,20,21,23,24]. Pericyte–endothelial cell associations were also enhanced by treatment with the viral proteins, individually, and even more so in combination. Although wounds treated with the mammalian proteins showed increased vascular density, the extent of pericyte coverage was less than with the viral combination. VEGF-A-treated wounds had poorer pericyte coverage than those treated with VEGF-E [17], while viral and mammalian IL-10 both appeared to enhance blood vessel maturation [29]. The effects of the VEGFs and IL-10s on wound bed expression of *Vegf-a* also mirrored their effects on blood vessel integrity, with a greater induction by the mammalian proteins relative to their viral homologues [17,29]. In future studies, it will be important to support these observations by examining VEGF-A protein levels in the healing wounds, and to investigate the effects of the viral proteins on VEGFR-2 activation and signaling in endothelial cells. Pericytes are mobilised and recruited to the newly formed vasculature by factors, such as TGFβ1, PDGF-β and the interferon-γ-inducible protein 10 (IP-10/CXCL10) [70,71,72], and directly inhibit endothelial cell proliferation and promote their differentiation, the result being increased blood vessel stability [73]. Differences in *Tgfβ1* and *Tgfβ3* levels were noted over time between the combination therapies. PDGF-β expression in murine wounds was also modulated by VEGF-A and VEGF-E treatment [17]. CXCL10 was also produced in response to the IL-10 activation of Stat1 in interferon-α-primed macrophages [74]. The altered expression of these regulators in wounds treated with the VEGF and IL-10 combinations is therefore highly likely to influence the quality of the new vasculature.

It is generally accepted that excessive inflammation, angiogenesis and fibrosis exasperates scar tissue formation [14,41,48,60]. Following the treatment of wounds with the viral proteins, individually and in combination, the residual scar was reduced. The viral combination dampened the formation of both the epidermal and dermal scar, while only a slight reduction in the dermal scar was observed with the mammalian combination. This is consistent with the ability of viral and mammalian IL-10s to reduce scar size [29,49,51], and the increased dermal scar observed in the presence of VEGF-A [16,17,75]. Here, treatment with VEGF-E, individually and in combination with ovIL-10, also enhanced dermal scar quality, while no benefits were seen with the mammalian combination. This reflects the reported ability of VEGF-A and VEGF-E administration to increasing the collagen content and improve basket weave-like bundle formation [17]. VEGF-A is thought to modulate collagen remodeling through the secretion of collagenases (MMP1, MMP3, and MMP13), and diminished expression of tissue inhibitors of MMPs [76,77]. These findings suggest that the viral protein combination has complimentary and beneficial effects on scarring, while the increase in dermal scarring induced by VEGF-A is not negated by IL-10. Critically, these scarring outcomes correlated with blood vessel quality in the treated wounds, as opposed to the level of inflammation or fibrosis.

Previous studies have indicated that the timing and route of administration greatly impacts the efficacy of wound healing therapies. VEGF-E and ovIL-10 have previously been administered subcutaneously to open wounds on horses during the initial inflammatory phase of healing [15]. This treatment regime promoted re-epithelialisation and dampened inflammation, but had no lasting effect on wound closure or blood vessel quality. The treatment of bandaged equine wounds with the viral proteins, administered after 24h with a topical hydrogel, also promoted re-epithelialisation and improved blood vessel quality [19]. This route of administration did not, however, impact wound inflammation, exuberant granulation tissue formation or the rate of closure. This indicated that longer term treatment was required, and that either route of administration was appropriate for VEGF-E, while ovIL-10 injection into or below the skin was necessary. In this murine model, subcutaneous injection of VEGF-E and ovIL-10 into the wound periphery until the point of closure was successful at improving both healing and scarring outcomes.

To our knowledge, no other combination wound healing therapies involving a growth factor and anti-inflammatory have been investigated. Others have explored the dual delivery of VEGF-A with either PDGF-β, fibroblast growth factor-4, insulin growth factor-1, keratinocyte growth factor, epidermal growth factor, angiopoietin-1 or bone morphogenic protein-2 on the healing of cutaneous wounds, hind-limb ischemia and bone defects [78,79,80,81,82,83]. The impact of a fusion between IL-10 and IL-4 has also been explored in the treatment of neuro-inflammatory pain [84]. Our findings demonstrate that successful skin repair may require a combination therapy that suppresses wound inflammation, while enhancing the proliferative and remodeling processes. This observation is consistent with findings from the germ-free adult mouse, in which the rapid and scar-free healing of skin wounds was associated with increased VEGF-A and IL-10 production [85], but differs from observations in fetal skin, for which regenerative healing is achieved in the presence of high IL-10 and low VEGF-A expression [50,75]. Here, VEGF-E and ovIL-10 were shown to act synergistically to promote skin repair, with complimentary effects on scar quality. This study did not, however, address whether the additive effects between VEGF-E and ovIL-10 specifically required the viral IL-10. It is possible that a combination of VEGF-E with mammalian IL-10, or any other anti-inflammatory, would show similar benefits. While the mammalian combination showed additive effects on healing, the IL-10 failed to counteract the pro-scarring effects of VEGF-A. With only a single-dose combination trialed, it is conceivable that alterations to the VEGF-A:IL-10 ratio would improve scarring outcomes.

Healing impairments greatly impact the quality of life for an estimated 37 million people globally suffering from chronic wounds [86], and an estimated 100 million surgical patients who develop excessive scarring [87]. Chronic wounds present with impaired re-epithelialisation, re-vascularisation and VEGF-A production, and persistent inflammation and IL-10 production [88,89,90]. Human trials also showed that the treatment of chronic wounds with recombinant VEGF-A can accelerate healing [91]. By contrast, hypertrophic and keloid scars are exasperated by chronic inflammation and abnormally low levels of IL-10, and show excess angiogenesis and VEGF-A production [92,93,94,95,96,97,98,99]. The treatment of human incisions with low concentrations of recombinant IL-10 improved scar appearance [25]. This evidence suggests that the relative abundance of endogenous VEGF-A and IL-10 in human wounds critically impacts both healing and scarring outcomes. Although our data were generated in uncomplicated cutaneous wounds of healthy mice, which, compared to human wounds, heal primarily by contraction with relatively little scarring, this study shows that the VEGF-E and ovIL-10 combination can target healing and scarring impairments associated with human skin wound complications.

## 5. Conclusions

This study demonstrated that the viral immunomodulators VEGF-E and ovIL-10, when applied to murine cutaneous wounds, act synergistically to enhance skin repair, and in a complimentary fashion to improve scar quality. The viral proteins augmented wound closure by promoting wound re-epithelialisation and re-vascularisation, and through the suppression of M1 macrophage and myofibroblast infiltration and retention of M2 macrophages, led to improved blood vessel stabilisation and collagen remodeling. The mammalian combination of VEGF-A and IL-10, while showing equivalent reparative benefits, differed from the viral combination in its ability to regulate TGFβ isoform expression, macrophage and pericyte recruitment, collagen remodeling and ultimately scar tissue area. These findings suggest that the viral proteins exhibit distinct therapeutic advantages over their mammalian counterparts. But more importantly, they support future research into combination therapies, be they viral or mammalian, that enable phasic targeting of the wound healing response in a manner that improves both healing and scarring outcomes for cutaneous wound indications. 

## Figures and Tables

**Figure 1 jcm-09-01085-f001:**
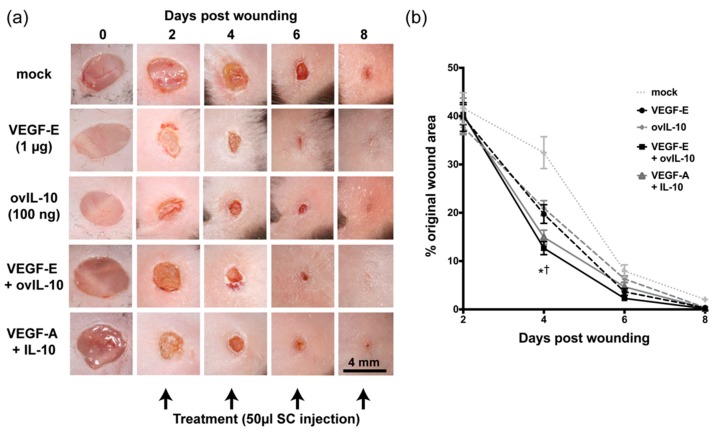
VEGF-E and ovIL-10 synergistically enhance wound closure. (**a**) Representative photographs of the healing process of 4 mm cutaneous full-thickness punch wounds following treatment with saline (mock), VEGF-E, ovIL-10, VEGF-E and ovIL-10, or VEGF-A and IL-10. Timing of treatments is indicated below, with dosages listed below the treatments. Scale bar = 4 mm. (**b**) Kinetics of wound closure in groups of mice following SC injection of the indicated treatment. Data are presented as the mean ± SEM; *n* = 8 wounds, and *p* values ≤ 0.05 are indicated when significant differences between mock-treated wounds and all other wound groups (*) or VEGF-E and ovIL-10 combination-treated and VEGF-E and IL-10 individually treated wounds (†) were identified.

**Figure 2 jcm-09-01085-f002:**
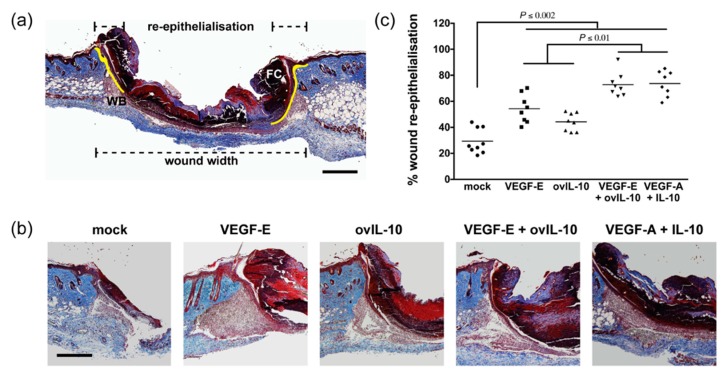
VEGF-E and ovIL-10 synergistically enhance wound re-epithelialisation. (**a**) Representative image of a day 3 wound section, stained with MSB trichrome. The neo-epidermis–dermis boundary is highlighted in yellow. Dashed lines indicate wound width, distance between cut edges of the panniculus carnosus, and coverage by the neo-epidermis. WB = wound bed; FC = fibrin clot. Scale bar = 300 µm. (**b**) Representative images of day 3 wound sections, showing the extent of re-epithelialisation 24 h after SC injection of the indicated treatment. Scale bar = 150 µm. (**c**) Percentage wound re-epithelialisation at day 3 post-wounding. Data are presented with each symbol representing an individual wound, and the line indicating the mean of 8 wounds, *p* values are indicated.

**Figure 3 jcm-09-01085-f003:**
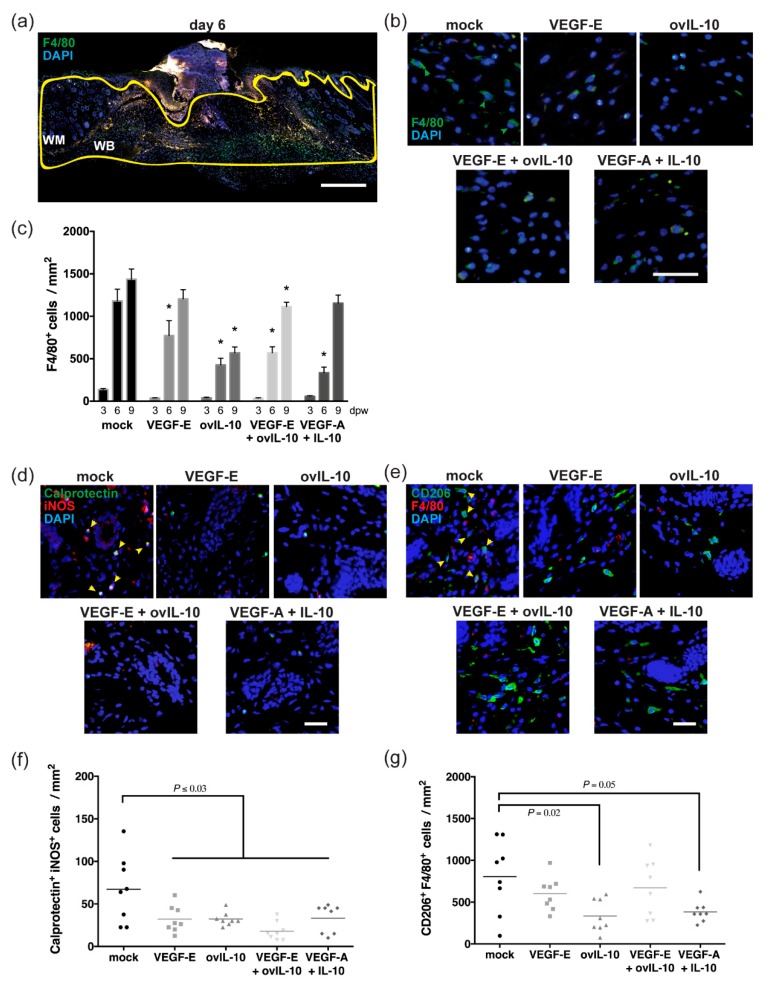
VEGF-E and ovIL-10 have complimentary effects on the response of macrophages to wounding. (**a**) Representative image of a day 6 wound section, stained with antibody to F4/80 (green) and the nuclear stain DAPI (blue). The wound bed (WB) and wound margin (WM) are outlined with a yellow line. Scale bar = 300 µm. (**b**) Representative images of the WM adjacent to the WB in day 6 wound sections after two SC injections of the indicated treatment. Stained macrophages indicated by green arrows. Scale bar = 50 µm. (**c**) Number of F4/80^+ve^ macrophages per mm^2^ in the WB and WM at 3, 6 and 9 days post-wounding (dpw). Data are presented as the mean ± SEM; n = 8 wounds, with means significantly different from that of mock-treated wounds indicated by an asterisk (*, *p* ≤ 0.05). Representative image of the WM in day 6 wound sections stained with antibodies to (**d**) calprotectin (green) and iNOS (red), or (**e**) CD206 (green) and F4/80 (red), and the nuclear stain DAPI (blue). Co-stained macrophages indicated by yellow arrows. Scale bar = 50 µm. Number of (**f**) calprotectin^+ve^ iNOS^+ve^ M1 macrophages and (**g**) CD206^+ve^ F4/80^+ve^ M2 macrophages per mm^2^ in the WB and WM at day 6 post-wounding. Data are presented with each symbol representing an individual wound, and the line indicating the mean of 8 wounds, *p* values are indicated.

**Figure 4 jcm-09-01085-f004:**
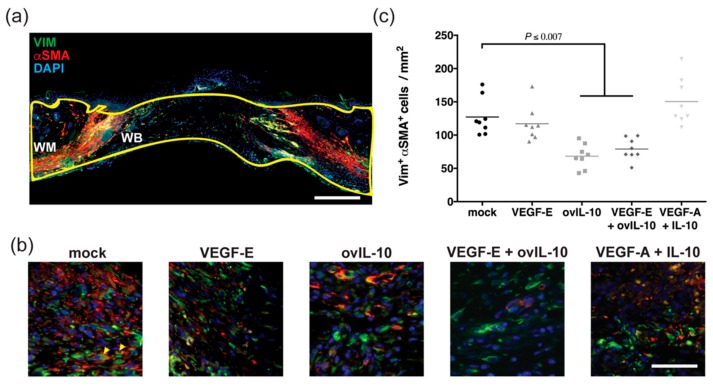
ovIL-10, with and without VEGF-E, reduces the response of myofibroblasts to wounding. (**a**) Representative image of a day 6 wound section stained with antibodies to vimentin (green) and αSMA (blue) and the nuclear stain DAPI (blue). The wound margin (WM) and wound bed (WB) are outlined with a yellow line. Scale bar = 300µm. (**b**) Representative images of the WM adjacent to the WB in day 6 wound sections after two SC injections of the indicated treatment. Co-stained myofibroblasts are indicated by yellow arrows. Scale bar = 50 µm. (**c**) The number of vimentin^+ve^ αSMA^+ve^ myofibroblasts within the WM and WB at day 6 post-wounding. Data are presented with each symbol representing an individual wound, and the line indicating the mean of 8 wounds. *p* values are indicated.

**Figure 5 jcm-09-01085-f005:**
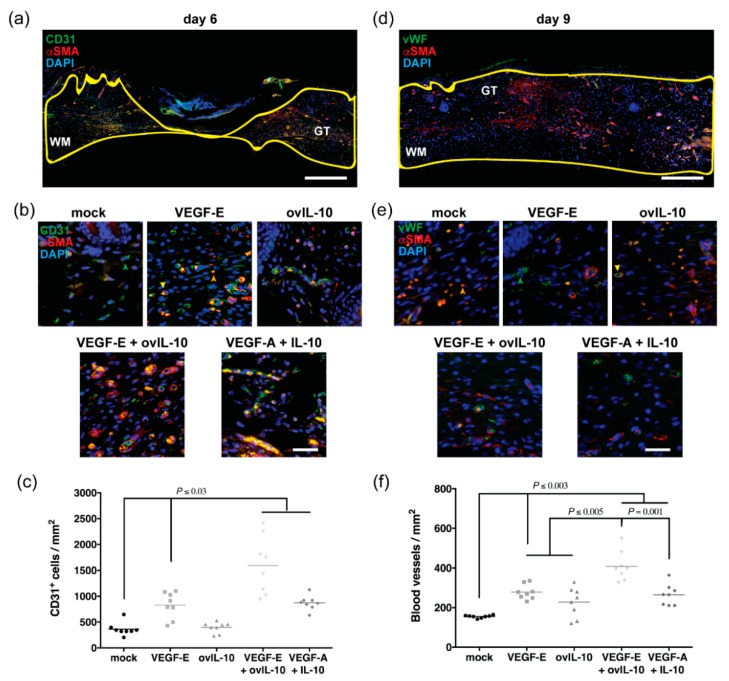
VEGF-E and ovIL-10 synergistically enhance wound re-vascularisation. (**a**) Representative image of a day 6 wound section stained with antibodies to CD31 (green) and αSMA (red) and the nuclear stain DAPI (blue). The wound margin (WM) and granulation tissue (GT) are outlined with a yellow line. Scale bar = 300 µm. (**b**) Representative images of the WM adjacent to the GT in day 6 wound sections after two SC injections of the indicated treatment. CD31^+ve^ endothelial cells and αSMA^+ve^ pericytes associated with the vasculature are indicated by green and yellow arrows, respectively. Auto-fluorescent red blood cells are indicated by orange arrows. Scale bar = 50 µm. (**c**) Number of CD31^+ve^ endothelial cells per mm^2^ in the WM and GT at day 6 post-wounding. Data are presented with each symbol representing an individual wound, and the line indicating the mean of 8 wounds, *p* values are indicated. (**d**) Representative image of a day 9 wound section stained with antibodies to vWF (green) and αSMA (red) and the nuclear stain DAPI (blue). The WM and GT are outlined with a yellow line. Scale bar = 300 µm. (**e**) Representative images of the GT near the upper WM in day 9 wound sections after four SC injections of the indicated treatment. vWF^+ve^ endothelial cells, αSMA^+ve^ pericytes and red blood cells are indicated by green, yellow and orange arrows, respectively. Scale bar = 50µm. (**f**) Number of αSMA^+ve^ pericyte-covered blood vessels per mm^2^ in the WM and GT at day 9 post-wounding. Data are presented with each symbol representing an individual wound, and the line indicating the mean of 8 wounds, *p* values are indicated.

**Figure 6 jcm-09-01085-f006:**
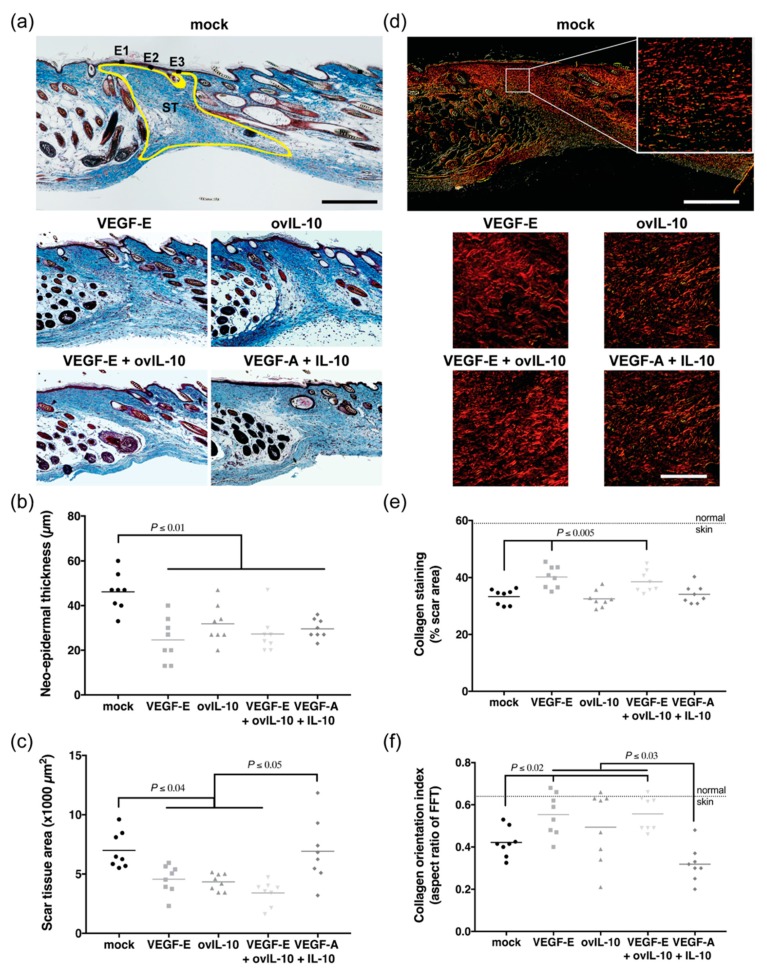
VEGF-E and ovIL-10 have complimentary effects on scar size and quality. (**a**) Representative images of MSB trichrome-stained day 16 wound sections after four SC injections of the indicated treatment. The sites of epidermal (E1–3) measurement are indicated and scar tissue area (ST) is outlined in yellow. Scale bar = 300 µm. (**b**) Thickness of the neo-epidermis in healed skin given the indicated treatments. (**c**) Scar tissue area in healed skin following the indicated treatments. Data are presented with each symbol representing an individual wound, and the line indicating the mean of 8 wounds, *p* values are indicated. (**d**) Representative image of picrosirius red-stained day 16 wound section visualised under dark-field with a polarised filter. Scale bar = 300 µm. Enlarged images show collagen density and orientation in ST adjacent to the neo-epidermis after four SC injections of the indicated treatment. Scale bar = 50 µm. (**e**) Percentage area of collagen staining in healed skin following the indicated treatment. (**f**) Collagen orientation indices (W/L ratio from zeroth-order maximum plot generated using FFT) in healed skin following the indicated treatments. Data are presented with each symbol representing an individual wound, and the line indicating the mean of 8 wounds, *p* values are indicated.

**Figure 7 jcm-09-01085-f007:**
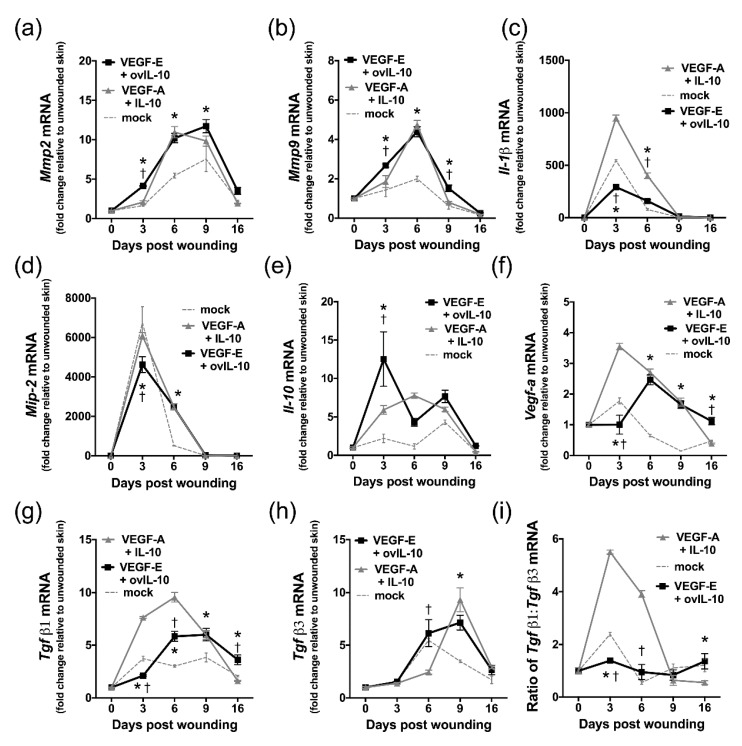
VEGF-E and ovIL-10 alter the expression of key wound healing mediators. Expression of (**a**) *Mmp2*, (**b**) *Mmp9*, (**c**) *Il-1β,* (**d**) *Mip-2*, (**e**) *Il-10*, (**f**) *Vegf-a*, (**g**) *Tgfβ1* and (**h**) *Tgfβ3* mRNA in wounds treated as indicated then harvested at day 0, 3, 6, 9 and 16 was determined by quantitative PCR. Data are presented as the mean ± SEM relative to GAPDH and unwounded skin; *n* = 4 samples, each containing 4 wounds. (**i**) Ratio of *TGFβ1* to *TGFβ3* mRNA levels in the wound for each treatment group, *p* values ≤ 0.05 are indicated when significant differences between VEGF-E and ovIL-10-treated and mock-treated wounds (*) or VEGF-E and ovIL-10-treated and VEGF-A and IL-10-treated wounds (†) were identified.

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
