# Peer review of "Orf Virus IL-10 and VEGF-E Act Synergistically to Enhance Healing of Cutaneous Wounds in Mice"

_jcm, 2020, doi:10.3390/jcm9041085_

Round 1
Reviewer 1 Report
The authors have made significant revisions to the manuscript. I feel that the revised manuscript provides the data to back the authors conclusions.
Reviewer 2 Report
This is a de novo submission to address the concerns previously highlighted in the original submission (jcm-555871). The additional experiments conducted by the authors now provide a full picture of the synergistic effects of VEGF-E and ovIL-10 combination treatment on healing and scarring processes in a murine wound healing model. I appreciate and acknowledge the authors significant efforts in addressing the previous concerns and have no further comments.
This manuscript is a resubmission of an earlier submission. The following is a list of the peer review reports and author responses from that submission.
Round 1
Reviewer 1 Report
The study of Wise et al. investigates the effects of recombinant Orf virus (ov) IL-10 and VEGF-E on skin wound healing. They compare single treatment as well as combinatory treatment of both viral proteins in comparison to recombinant mouse IL-10 and VEGF-A and find that ovIL-10 in combination with VEGF-E enhances wound closure, while the inflammatory response and fibrosis are significantly reduced.
The aim of the study is clearly stated and the report is well written and easy to follow. The material and method section provides detailed information on the experimental set up and analysis.
Minor comments:
1) Could the authors please comment on the rationale behind the protein quantity that is administered. Is this based on previous experiments? If so it would be interesting for the reader to know why the treatment is 1 µg of VEGF-E or VEGF and 0.1 µg ovIL-10 (or IL-10).
2) The results would benefit if data was shown as dot plots (one dot per one wound) instead of bar charts.
3) Figure 4 b: Mouse epidermis in unwounded skin is not different among individuals and therefore there is no need to calculate an Index. For clarity and simplicity, I suggest to measure the epidermal thickness above the scar tissue and present the data as “neo-epidermal thickness in µm”.
4) Figure 4 c: There is no gain for the calculation of a dermal index. For clarity and simplicity, I suggest to measure the area of scar tissue (as shown in a) in yellow) and present the data as “area of scar tissue in µm2”.
5) Figure 5 a-h: Please state unit on y-axis eg ‘fold increase over unwounded skin’
6) Citation 29 (line 580) and 33 (590) are same, please delete one.
Reviewer 2 Report
see attached file

Reviewer 3 Report
The manuscript by Wise et al. explored the synergistic effects of VEGF-E and ovIL-10 combination treatement on healing and scarring processes in a murine wound healing model. The dual combination was compared to single combinations of either VEGF-E or ovIL-10 alone. Additionally, a mammalian combination of VEGF-A and IL-10 was also used as another treatment group. Overall, this is a well-presented study and the authors have made a significant effort in understanding the effects of VEGF-E and ovIL-10 across numerous phases in wound healing, however there are several concerns that need to be addressed:
1. Can the authors clarify the selection of timing for the various immunohistochemical ex vivo analysis? The study had 4 different time points (3, 6, 9 and 16 days) in which re-epithelialisation was measured in day 3 wounds, dermal cell infiltrates measured in day 6 wounds, revascularisation measured in day 9 wounds and scar size/quality assessed in day 16 wounds. The authors should also assess macrophage accumulation and re-vascularisation at earlier timepoints (day 3 for macrophages and day 6 for neovascularisation) to see if the treatments are also effective at those timepoints. While the authors have nicely shown interesting findings at the selected timepoints, the differences of the treatments might be more striking at the earlier timepoints.
2. The authors have selected to use αSMA as both a myofibroblast and pericyte marker. αSMA is also often used to detect smooth muscle cells or as an indicator of more mature arteriolar formation. While αSMA is an established marker for both, the study would be strengthened if the authors also included an additional complementary marker to confirm the presence of myofibroblasts (e.g. vimentin, amine oxidase, copper containing 3, AOC3) and pericytes (e.g. NG-2, PDGFR-B).
3. Are the authors able to delineate whether the effect of VEGF-E+ovIL-10 treatment has any effect on macrophage polarisation towards a pro- or anti-inflammatory phenotype? This would be of significant interest given that the pro-inflammatory macrophages would be important in the early stages of wound healing, while the anti-inflammatory macrophages are critical in the resolution phase.
4. Are there any differences in the rate of wound closure across the treatment groups?
5. The authors have included a representative image showing the stained sections for Figures 1 – 4. It would be great if the authors also included representative images of the 5 different treatment groups. It does not have to be of the entire wound, but relevant areas/sections would be recommended.
6. The authors have shown interesting gene expression data for the combination treatments. Did the authors also measure the individual treatments (VEGF-E and ovIL-10) that were included for the earlier figures?
7. It would be interesting if the authors are also able to show corresponding protein changes at the timepoints where gene expression levels are most striking. For example, protein changes in VEGF-A at days 6/9 or MMP2 and MMP9 at day 6 could be explored.
8. VEGF-A exerts pro-angiogenic effects on the activation and phosphorylation of VEGFR2 and downstream activation of other signalling pathways including p38 MAPK, ERK and eNOS. Given the striking mRNA changes on VEGF-A expression, have the authors explored whether the treatments also influence the downstream mediators of VEGF-A?
9. The authors noted in their discussion that the TGF-β1/TGF-β3 expression ratio differed between the combination treatments. This should be included as a result panel within Figure 5.
10. Did the treatments influence expression changes of cell adhesion molecules (ICAM-1, VCAM-1, integrin β2 and L-selectin)?
11. Gene symbols for mice are normally italicised, first letter upper case all the rest lower case (e.g. Mmp2).
12. There are some minor typographical errors throughout the manuscript:
a. Line 157 – ref #33 is repeated multiple times, likely to just be an EndNote error.
b. Line 163 – gene is missing for the primer sequence
c. Line 335 – This was tested in a murine full-thickness…
d. Line 416 - … poorer pericyte coverage than those treated with…
Round 2
Reviewer 2 Report
Authors have made the necessary corrections.
Reviewer 3 Report
While the authors have addressed some of my earlier comments, they have not addressed most of them, citing the rapid deadline for resubmission. Respectfully, while the authors have attempted to overcome this by citing additional study limitations in the discussion, I strongly believe that these should be addressed in order to further complement this study. In the past, I believe journals in this publishing company will allow resubmissions at a later date in which the same reviewers assess the manuscript. The following concerns still need to be addressed:
The authors should assess macrophage accumulation and re-vascularisation at earlier timepoints (day 3 for macrophages and day 6 for neovascularisation) to see if the treatments are also effective at those timepoints. Given that they hypothesise that the combination therapy would accelerate these processes, it is justified that the earlier timepoints also be assessed. The authors should include additional complementary markers to confirm the presence of myofibroblasts (e.g. vimentin, amine oxidase, copper containing 3, AOC3) and pericytes (e.g. NG-2, PDGFR-B). The authors should delineate whether the effect of VEGF-E+ovIL-10 treatment has any effect on macrophage polarisation towards a pro- or anti-inflammatory phenotype? This would be of significant interest given that the pro-inflammatory macrophages would be important in the early stages of wound healing, while the anti-inflammatory macrophages are critical in the resolution phase. The authors should show corresponding protein changes at the timepoints where gene expression levels are most striking. For example, protein changes in VEGF-A at days 6/9 or MMP2 and MMP9 at day 6 could be explored. Did the treatments influence expression changes of cell adhesion molecules (ICAM-1, VCAM-1, integrin β2 and L-selectin)? Gene symbols for mice are normally italicised, first letter upper case all the rest lower case (e.g. Mmp2). There are some minor typographical errors throughout the manuscript. These were not addressed in the revised version: Line 163 – gene is still missing for the primer sequence Line 375 – This was tested in a murine full-thickness… Line 466 - … poorer pericyte coverage than those treated with…